# Prospective pilot study on the predictive significance of plasma miR-30b-5p through the study of echocardiographic modifications in Cavalier King Charles Spaniels affected by different stages of myxomatous mitral valve disease: The PRIME study

**Sara Ghilardi**[1], **Cristina Lecchi**[1], **Mara Bagardi**[1]*, **Giovanni Romito**[2], **Fabio M. Colombo**[3], **Michele Polli**[1], **Celeste Franco**[1], **Paola G. Brambilla**[1]

1 Department of Veterinary Medicine and Animal Science, University of Milan, Lodi, Italy, 2 Department of Veterinary Medical Sciences, Alma Mater Studiorum–University of Bologna, Bologna, Italy, 3 Department of Veterinary Sciences for Health, Animal Production and Food Security–Lodi (Italy), University of Milan, Milan, Italy

* mara.bagardi@unimi.it, mara.bagardi@anicura.it

## Abstract

Specific microRNAs expressions may accurately characterize different stages of canine myxomatous mitral valve disease. This preliminary pilot study aimed to (1) describe the clinical and echocardiographic parameters of Cavalier King Charles Spaniels affected by myxomatous mitral valve disease at different American College of Veterinary Internal Medicine (ACVIM) stages (B1, B2 and C) and healthy control group (ACVIM A), comparing the parameters collected during the first examination (T0) and the end of the follow-up (T1); (2) assess the association between the values of echocardiographic parameters at T1 and the expression profile of miR-30b-5p at T0. Thirty-five Cavalier King Charles Spaniels (median age 4.29 years and median weight 9 Kg) in different ACVIM stages were included (7 A, 19 B1, 6 B2 and 3 C). Inverse probability weighting analysis was performed to estimate the association of the exposure variable (miR-30b-5p) with the outcome variables (clinical and echocardiographic variables). Time was included as variable. The results pointed out that high levels of plasma miR-30b-5p corresponded to lower values of left ventricular end-diastolic diameter normalized for body weight, end-diastolic and end-systolic volumes indexed for body weight, and left atrium-to aortic root ratio. Hence, higher miR-30b-5p expressions were associated with milder forms of mitral valve disease in our study population. In contrast, the results obtained for the intensity of heart murmur, the mitral regurgitation severity, and the Mitral INsufficiency Echocardiographic score) were not statistically significant. A relationship between high abundance of miR-30b-5p and myxomatous mitral valve disease that appear echocardiographically more stable over time has been demonstrated. In conclusion, Cavalier King Charles Spaniels affected by myxomatous mitral valve disease that at the first cardiologic evaluation showed an

**Data Availability Statement:** All relevant data are within the paper and its Supporting information files.

**Funding:** The authors received no specific funding for this work.

**Competing interests:** The authors have declared that no competing interests exist.

upregulation of miR-30b-5p are expected to experience lesser variations on their echocardiographic examination between T0 and T1.

## Introduction

Myxomatous mitral valve disease (MMVD) is the most common cardiovascular disease in dogs and a frequent cause of heart failure [1,2]. Despite its large diffusion, the underlying causes and mechanisms involved in the development of valve degeneration and disease progression remain to be conclusively elucidated [1,2]. Although a causative mutation has not been identified yet, canine MMVD is reported to be hereditable in some breeds, including the Cavalier King Charles Spaniel (CKCS). Approximately 100% of CKCSs develop MMVD at the age of 10 years, and a polygenic mode of inheritance has been suggested [3–5]. The application of advanced genomic technologies has potential to provide information that may impact treatment, prevalence, or severity of MMVD through the elucidation of pathogenic mechanisms and the detection of predisposing genetic loci of major effect [6].

Transthoracic echocardiography is currently considered the gold standard for the diagnosis of MMVD and the evaluation of the follow-up of the affected dogs [2]. However, this test needs specialized equipment and well-trained veterinarians to reduce interobserver variability, especially in the early stages of the disease as valves affected by mild changes may be misinterpreted as normal.

The identification of reliable biomarkers may enhance the robustness of clinical decision-making and may pave the way for the development of alternative/complementary diagnostic strategies [7]. In recent years, the study of microRNAs (miRNAs)as promising diagnostic and prognostic biomarkers for the detection of disease increased [8]. They are short non-coding RNA molecules (21–25 nucleotides in size) found in plants, animals and some viruses, that regulate post-transcriptional silencing of target genes involved in pathophysiologic conditions in many mammalian species [9–15], including cardiovascular diseases [16,17]. In human medicine, plasma and serum miRNAs are currently being intensively investigated and specific miRNA expression patterns have been reported for various cardiovascular diseases [17]. In contrast, in veterinary medicine, the list of studies on circulating miRNAs is currently brief and few of them concern canine MMVD [6,18–22]. Specifically, Hulanicka et al., in 2014 [18], analysed the expression of 9 specific miRNAs in the plasma of Dachshunds suffering from MMVD. Enrolled dogs were classified adapting the American College of Veterinary Internal Medicine (ACVIM) classification as stage A (i.e., unaffected control dogs; n. 8), B (asymptomatic dogs, considering together the ACVIM stages B1 [i.e., dogs without cardiac enlargement] and B2 [dogs with cardiac enlargement]; n. 8) and C (considering together the ACVIM stages C [i.e., dogs that have experienced an episode of congestive heart failure] and D [i.e., dogs affected by refractory congestive heart failure]; n. 8). Expression analysis using the Real-Time Quantitative Reverse Transcription PCR (RT-qPCR) method revealed that two of these miRNAs were significantly downregulated (i.e., the expression of miR-30b and miR-133b differed between stage B and A and between stage C and A, respectively) [18]. In 2015, Li and co-workers [20] investigated a larger number of miRNAs (n. 277) by RT-qPCR in dogs belonging to various breeds and affected by MMVD at different ACVIM stages (A = 6; B1+B2 = 6; C+D = 6). Stage B1+B2 and C+D showed 11 dysregulated miRNAs (four upregulated and seven downregulated) as compared to stage A. Moreover, the expression of six out of 11 miRNAs was significantly different between stages B1+B2 and C+D [20]. In 2017, Yang and colleagues [22] for the

first time evaluated miRNAs from exosomes (i.e., extracellular vesicles of endosomal origin, enriched in cardioprotective miRNAs, which behave as key mediators of intercellular communication) using RT-qPCR in 13 healthy dogs and 14 affected by MMVD at various stages (asymptomatic = 7; symptomatic = 7). They demonstrated that exosomal miR-181 and miR-495 significantly increased in dogs with symptomatic MMVD compared to the others, whereas miR-9 and miR-599 increased and decreased, respectively, in all MMVD-affected dogs compared to healthy dogs. This led to the hypothesis that exosomal miRNA expression-level changes may be more specific to the disease status than total plasma miRNAs [22]. However, processing of exosomal miRNAs is not standardized yet, so a better consistency in the sample processing is still needed. For this reason, quantification of plasma miRNAs is preferable, at present [23]. In 2018, the expression of circulating miRNAs via genome-wide sequencing through the enrolment of 17 dogs of various breeds (8 dogs with congestive heart failure secondary to MMVD and 9 age-matched healthy dogs) was profiled by sequencing and qPCR validation [19]. The analysis showed the up-regulation of four (miR-133, miR-1, miR-let-7e, and miR-125) and the downregulation of four miRNAs (miR-30c, miR-128, miR-142, and miR-423) in MMVD-affected dogs [19]. In 2021, Ro et al. [21] used RT-qPCR to evaluate levels of 11 miRNAs in 82 dogs of various breeds, including 10 healthy dogs and 72 dogs with different heart diseases (including 35 affected by MMVD: stage B1 = 1; B2 = 7; C = 22; D = 5). In dogs affected by MMVD, upregulation of cfa-miR-130b was documented [21]. In 2022, Bagardi et al. [7] measured the concentration of miR-1-3p, miR-30b-5p and miR-128-3p in 33 CKCSs of ACVIM stage B1 divided into three age-related groups, and 11 ACVIM stage A CKCSs. The results of this study showed that only miR-30b-5p was significantly higher in ACVIM B1 dogs than in ACVIM A subjects. According to the age of dogs, the amount of miR-30b-5p was significantly higher in group B1<3y, B1 3-7y, and B1>7y than in group A. The areas under the receiver operating curves were fair in discriminating between group B1<3y and group A, and between B1 3-7y and A, and good in discriminating between group B1>7y and A [7]. This study demonstrates that the identification of dogs with early asymptomatic MMVD through the evaluation of miR-30b-5p represents an intriguing possibility that certainly merits further research.

Although these results could allow speculating that specific miRNAs expressions may accurately characterize different stages of MMVD, further research is necessary to confirm such a hypothesis, especially considering the remarkable limitations of the abovementioned studies (i.e., a limited sample size of each research, potential confounding effects associated with the heterogeneous breeds, age categories, MMVD stages as well as types of evaluated miRNAs, and lack of follow-up). Thus, the need arises to create studies that are statistically and methodologically more robust. To achieve this, the present work tries to assess the predictive value of one of the most promising of the miRNAs investigated to date [7,18], namely miR-30b-5p, by correlating its expression to the clinical and echocardiographic parameters of CKCSs affected by MMVD at different ACVIM stages. The choice to focus the attention exclusively on this miRNA has been dictated by the crossed findings reported in literature for the Dachshund by Hulanicka et al. [18] and for the CKCS by Bagardi et al. [7]. Both works, in fact, reported a significant downregulation of miR-30b-5p in MMVD-affected dogs belonging to ACVIM classes B (asymptomatic in general, with and without cardiac remodelling) and B1, respectively, when compared to healthy subjects.

The present study aims at (1) describing the clinical and echocardiographic parameters of healthy CKCSs (ACVIM A) and CKCSs affected by MMVD at different ACVIM stages (B1, B2 and C), comparing the obtained parameters at T0 (first examination) and T1 (end of the follow-up); and (2) assessing the association between the values of clinical and echocardiographic parameters at T1 and the expression profile of miR-30b-5p measured at T0.

## Materials and methods

### Clinical and echocardiographic examinations

This is a preliminary prospective pilot study. Owned CKCSs visited at the Cardiology Unit of the Department of Veterinary Medicine, University of Milan, between November 2019 and September 2021 were considered for enrollment. Informed consent was signed by the owners, according to the ethical committee statement of the University of Milan, number 2/2016.

Every dog was evaluated on two different occasions, a first examination (T0), and a follow-up clinic visit (T1). For this study, the time between T0 and T1 had not to be standardized but could vary among the recruited patients between 6 and 24 months.

At T0, animal history was collected and a complete physical examination was performed. The cardiovascular system was evaluated through cardiac auscultation performed by trained cardiologists (MB and PGB), who established the presence/absence of a cardiac murmur and its timing and intensity. Cardiac murmurs were classified as follows: 0 = absent; 1 = I-II/VI left apical systolic or soft; 2 = III-IV/VI bilateral systolic or moderate and loud, respectively; 3 = V-VI/VI bilateral systolic or palpable [24]. Blood pressure was indirectly measured through an oscillometric method according to the ACVIM consensus statement [25].

A standard transthoracic echocardiographic examination was performed by a single operator (MB) to detect MMVD using MyLabOmega Vet ultrasound machine (Esaote, Genova, Italy), equipped with multi-frequency phased array probes (1–5 and 2–9 MHz). Probes were chosen according to the weight of the subject. Echocardiographic examinations were performed with continuous electrocardiographic monitoring and video clips were obtained to make offline measurements [26]. Dogs were gently restrained during the examinations and were classified according to the ACVIM classification scheme [27].

The echocardiographic parameters considered pivotal to the MMVD diagnosis and monitoring over time and included in the study are described in Table 1.

At the end of the echocardiographic examination, the Mitral INsufficiency Echocardiographic (MINE) score was calculated by assigning a score from 1 to 4 to the left ventricular end-diastolic diameter normalized for body weight (LVIDdN), left atrial-to-aortic root ratio (LA/Ao), E wave peak velocity (E) and fractional shortening (FS%) values. Every patient was classified on the base of the total reached score, as follows: mild (total score = 4–5), moderate (total score = 6–7), severe (total score = 8–12) and late stage (total score = 13–14) [32].

Peripheral venous blood sampling was performed at the end of the echocardiographic examination. Dogs were fasted for at least 12 hours. Blood was collected from the cephalic or saphenous vein in two 2.5-mL EDTA tubes and one serum collection tube. Blood from one EDTA tube was used for a complete blood count, while blood from the serum collection tube was used for biochemical analysis. The leftover was used for plasma miRNAs extraction. According to the study design, at T1, every procedure described for T0 was repeated, except for plasma miR-30b-5p extraction. Every variation of animal history, clinical signs, and echocardiographic parameters was carefully noted to monitor MMVD progression since T0.

CKCSs with other systemic diseases such as systemic hypertension, primary pulmonary hypertension, neoplasia, and other cardiac abnormalities such as dilated cardiomyopathy, congenital cardiac abnormalities, endocarditis, and severe arrhythmia were excluded from the study.

### MiR-30b-5p extraction and RT-qPCR quantification

SmallRNA extraction was performed as previously reported for samples collected at T0 [7]. Briefly, within two hours from sampling, the leftover blood was centrifuged at 800 g for 15 minutes. Plasma was then stored at -80˚C until RNA isolation.

**Table 1. List of the echocardiographic parameters analyzed in this study, and description of how they were obtained.**

| Parameter | Description | Reference |
|---|---|---|
| LVIDdN | M-mode measurement of the left ventricle end-diastolic diameter obtained from a right parasternal short-axis view at the level of the papillary muscles, normalized for body weight | [28] |
| LA/Ao | Two-dimensional (2D) measurement obtained from a right parasternal short-axis view at the level of the base of the heart | [29] |
| E | Obtained from a left parasternal long-axis 4-chamber view through a pulsed-wave Doppler mode | [30] |
| ESVI and EDVI | Left ventricular end-systolic and end-diastolic volume index, respectively, obtained from a 2D-guided M-mode echocardiogram in a right parasternal short-axis view at the level of the papillary muscles and divided by the body surface area of the patient | [31] |
| FS% | Measurement obtained from a 2D-guided M-mode echocardiogram of the left ventricle from a right parasternal short-axis view at the level of the papillary muscles according to the following formula: [(LVIDd—LVIDs) / LVIDd] x 100 | [30] |
| Mitral regurgitant jet size | Assessed in a semi-quantitative way by calculating the maximal ratio of the regurgitant jet area signal to the left atrial area using color-flow Doppler mode. Mitral regurgitation is considered trivial or trace if such a ratio is $< 10\%$, mild if it is $\geq 10\%$ but $< 20\text{–}30\%$, moderate if it is $\geq 20\text{–}30\%$ but $< 70\%$, and severe if it is $\geq 70\%$. Specifically, mitral regurgitation is considered as trivial when it is not always visible during all systolic cycles, while it is considered as trace when it is always detectable | [30] |

E = E wave peak velocity, ESVI = end-systolic volume index, EDVI = end-diastolic volume index, FS% = fractional shortening, LA/Ao = left atrial-to-aortic root ratio, LVIDd = left ventricular end-diastolic diameter, LVIDdN = left ventricular end-diastolic diameter normalized for body weight, LVIDs = left ventricular end-systolic diameter.

An aliquot of 150 μL of plasma per sample was thawed on ice and centrifuged at $3000 \times g$ for 5 min at 4 ˚C and smallRNA was extracted using miRNeasy Serum/Plasma Kits (Qiagen, Cat. No. 217184, Milano, Italy) following the manufacturer's instructions. The sample was incubated at room temperature for 5 min and 3.75μL (25 fmol final concentration) of the synthetic spike-in control *Caenorhabditis elegans* miRNA cel-miR-39 (Qiagen, Cat. No. 219610) was spiked into samples. RNA extraction was then carried out according to the manufacturer's instructions. RNA concentration was assayed using a NanoDrop ND-1000 spectrophotometer. Reverse transcription was carried out on 10 ng of total RNA using a TaqMan Advanced miRNA cDNA Synthesis Kit (Cat. No. A28007, Applied Biosystems) following the manufacturer's instructions.

RT-qPCR was performed following the MIQE guidelines [33]. The small RNA TaqMan assays (ThermoFisher Scientific) included cel-miR-39-3p (assay ID 478293_mir) and miR-30b-5p (assay ID 478007_mir). The reference miRNA was miR-16-5p (assay ID rno481312_-mir). Quantitation was performed in 15 μl in a CFX Connect Real-Time PCR Detection System (Bio-Rad) as previously reported [33]. No-RT controls and no-template controls were included. MiRNA expressions are reported in terms of fold change normalized to miR-16 using the formula $2^{-\Delta\Delta Cq}$ on Bio-Rad CFX Maestro Software.

## Statistical analysis

Statistical analysis was performed using commercially available statistics software (SPSS® 27.0, IBM®, SPSS, USA). Descriptive statistics were generated. The distribution of data for continuous variables was assessed for normality using the Shapiro-Wilk test. Normally distributed parameters are expressed as mean ± standard deviation, while non-normally distributed

parameters are reported as the median and interquartile range (25[th] and 75[th] percentile). Obtained values were defined as statistically significant at a *P* value < 0.05, and all the significance values were adjusted according to the Bonferroni post hoc correction.

The Inverse Probability Weighting (IPW) method was performed to face confounding biases using "R" software [34] to estimate the association of the exposure variable (miR-30b-5p) with the outcome variables (clinical and echocardiographic variables) considering the time. The regressors were miR-30b-5p, time, and their interaction. A gaussian model was used for continuous responses (LA/Ao, EDVI, ESVI, LVIDdN, FS% and E) and an ordinal model was used for ordinal responses (presence/intensity of a heart murmur, mitral regurgitant jet size, MINE score). For each outcome model and each response variable, the marginal effect of the exposure variable (miR-30b-5p) and its standard error were calculated. The marginal effect of miR-30b-5p for continuous responses was interpreted as the increase/decrease (depending on the positive or negative marginal effect value) of the response variable, corresponding to an increase of miR-30b-5p in one unit. The marginal effect of miR-30b-5p for ordinal responses was interpreted as the increase/decrease of the probability of the lowest-value category when miR-30p-5p increases by one unit [35].

## Results

### Demographics and characteristics of the study population

The study population was composed of 35 CKCS, of which 23 (65.7%) were females and 12 (34.3%) males. At T0, the median age was 4.29 years (2.07–7.66), and the body weight was 8.99 ± 2.08 kg. Clinical and echocardiographic characteristics and plasma miR-30b-5p expression of the population are summarized in Table 2.

All the enrolled dogs had no abnormalities on complete blood count and biochemical analysis. All dogs were re-evaluated at a follow-up clinic visit (T1), although the time between T0 and T1 varied among the recruited patients as follows: between 6 and 12 months (14 dogs), between 12 and 18 months (19 dogs), and between 18 and 24 months (2 dogs). The ACVIM stage of some dogs evolved over time. Specifically, 5 of the 7 dogs that had been classified as ACVIM class A at T0 were classified as a class B1 at T1, all the B1 subjects remained in the same ACVIM class, one ACVIM class B2 dog at T0 was classified as a class C at T1, and one ACVIM class C dog at T0 was classified as a class D at T1. Similarly, also some demographic, clinical and echocardiographic variables changed from T0 to T1, as summarized in Table 3.

### IPW analysis for continuous variables

According to the IPW analysis, the obtained results were statistically significant for 4 of the selected echocardiographic parameters, namely LA/Ao, ESVI, EDVI, and LVIDdN. Specifically, the marginal effect of miR-30b-5p was negative for each of these parameters, which means that any increase in one unit of plasma miR-30b-5p expression corresponded to a decrease of the echocardiographic variable in a number denominated as "estimate". In particular, the marginal effect of miR-30b-5p was as follows: LVIDdN (*estimate*: -0.0004, *P* = 0.0003), ESVI (*estimate*: -0.0124, *P* = 0.005), EDVI (*estimate*: -0.0503, *P* = 0.0005), LA/Ao (*estimate*: -0.0004, *P* = 0.048) (S1 Table).

Fig 1 shows the marginal effect of plasma miR-30b-5p expression on each echocardiographic variable at T0 and T1. For a better understanding of the figure, it should be considered that when T0 and T1 lines are less parallel, the tendency to an interaction effect between miR-30b-5p and the echocardiographic parameter is higher (i.e., ESVI).

Through the IPW analysis we also studied the marginal effect of the elapsed time between T0 and T1 on the echocardiographic variables. Interestingly, none of the results reached the

**Table 2. Clinical and echocardiographic data, and plasma miR-30b-5p expression of the CKCS population divided according to the ACVIM classification scheme.** Data obtained at T0.

| | Overall population | A | B1 | B2 | C |
|---|---|---|---|---|---|
| **Number of dogs** | 35 | 7 | 19 | 6 | 3 |
| **Sex** | 23F (5NF) 12M | 6F (1NF) 1M | 14F (2NF) 5M | 2F (1NF) 4M | 1F (1NF) 2M |
| **Age (years)** | 4.29 (2.07–7.66) | 2.07 (1.65–3.23) | 3.46 (2.02–7.38) | 8.32 (6.89–11.13)[a,b] | 8.80 (8.22–9.20)[a,b] |
| **Weight (kg)** | 9.00 (7.50–10.35) | 7.80 (7.20–8.00) | 9.10 (7.40–9.75) | 10.67 (8.02–12.57) | 10.40 (9.70–12.67) |
| **Murmur** | 16 grade 0 9 grade 1 7 grade 2 3 grade 3 | 7 grade 0 | 9 grade 0 9 grade 1 1 grade 2 | 5 grade 2 1 grade 3 | 1 grade 2 2 grade 3 |
| **Regurgitant jet size** | 7 grade 0 10 grade 1 3 grade 2 6 grade 3 7 grade 4 2 grade 5 | 7 grade 0 | 10 grade 1 3 grade 2 5 grade 3 1 grade 4 | 1 grade 3 4 grade 4 1 grade 5 | 2 grade 4 1 grade 5 |
| **LVIDdN (cm/kg)** | 1.44 (1.29–1.60) | 1.31 (1.23–1.45) | 1.35 (1.23–1.48) | 1.89 (1.76–2.12)[a,b] | 2.07 (2.01–2.21)[a,b] |
| **ESVI ml/m²** | 21.41 (14.67–28.98) | 14.67 (10.72–18.20) | 19.01 (13.51–22.76) | 38.79 (31.07–46.73)[a,b] | 36.96 (32.97–39.29)[a,b] |
| **EDVI ml/m²** | 64.29 (48.33–83.45) | 50.50 (42.96–64.29) | 53.86 (43.25–69.37) | 123.15 (104.35–163.43)[a,b] | 156.58 (145.41–182.74)[a,b] |
| **LA/Ao** | 1.23 (1.08–1.49) | 1.34 (1.03–1.43) | 1.17 (1.04–1.30) | 1.90 (1.80–2.08)[a,b] | 1.82 (1.81–1.92)[a,b] |
| **E (m/s)** | 0.75 (0.67–0.83) | 0.71 (0.61–0.80) | 0.68 (0.64–0.79) | 0.99 (0.77–1.29)[b] | 1.44 (1.42–1.48)[a,b] |
| **FS (%)** | 37.00 (31.00–42.00) | 40.00 (34.00–45.00) | 33.00 (28.00–39.00) | 38.50 (34.75–46.50) | 48.00 (44.50–49.00)[b] |
| **MINE score** | 28 mild 3 moderate 4 severe | 7 mild | 19 mild | 2 mild 2 moderate 2 severe | 1 moderate 2 severe |
| **miR-30b-5p (fold change)** | 24.58 (11.46–66.68) | 14.43 (1.73–25.54) | 29.57 (9.82–168.72) | 24.22 (16.73–92.86) | 34.81 (26.25–197.63) |

E = E wave velocity, ESVI = end-systolic volume index, EDVI = end-diastolic volume index, FS = fractional shortening, LA/Ao = left atrial-to-aortic root ratio, LVIDdN = left ventricular end-diastolic diameter normalized for body weight.

Murmur = systolic heart murmur intensity: 0 = absent, 1 = I-II/VI left apical systolic, or soft, 2 = III-IV/VI bilateral systolic, or moderate and loud, 3 = V-VI/VI bilateral systolic or palpable.

Sex: F = female, NF = neutered female, M = male.

Regurgitant jet size: 0 = absent, 1 = trivial, 2 = trace, 3 = mild, 4 = moderate, 5 = severe.

All data are expressed as median and interquartile range (25th and 75th percentile).

[a]Values within a row differ significantly at $P < 0.05$ from ACVIM class A subjects.

[b]Values within a row differ significantly at $P < 0.05$ from ACVIM class B1 subjects.

statistical significance: LVIDdN ($P = 0.194$), ESVI ($P = 0.534$), EDVI ($P = 0.121$), LA/Ao ($P = 0.602$), FS% ($P = 0.074$), E ($P = 0.785$).

## IPW analysis for ordinal variables

As previously explained, the IPW analysis tried to determine whether plasma miR-30b-5p expression has a marginal effect even on ordinal variables of our choice, specifically the presence/intensity of a heart murmur, the mitral regurgitant jet size, and the MINE score. None of the abovementioned parameters showed a statistically significant result when analyzed along the plasma miR-30b-5p expression; however, the marginal effect of the miRNA on these variables is always positive (S1 Table). Since these variables have been expressed in different classes to determine their severity, the positive marginal effect of miR-30b-5p examined in the IPW analysis means that any increase in one unit of plasma miR-30b-5p expression corresponds to an increase of the probability of the class with the lower value. Specifically, these classes are:

**Table 3. Clinical and echocardiographic data of the CKCS population divided according to the ACVIM classification scheme.** Data obtained at T1.

| | Overall population | A | B1 | B2 | C | D |
|---|---|---|---|---|---|---|
| Number of dogs | 35 | 2 | 24 | 5 | 3 | 1 |
| Sex | 23F (5NF) 12M | 2F | 18F (3NF) 6M | 2F (1NF) 3M | 1F (1NF) 2M | 1M |
| Age (years) | 5.69 (3.45–8.84) | 3.59 (2.63–4.54) | 4.47 (3.02–7.49) | 9.94 (8.29–12.48) | 9.27 (8.78–9.69) | 8.26 |
| Weight (kg) | 8.50 (7.60–10.60) | 8.05 (7.60–8.50) | 8.50 (7.60–9.65) | 7.40 (6.45–12.60) | 10.60 (9.90–12.05) | 10.80 |
| Murmur | 13 grade 0 11 grade 1 8 grade 2 3 grade 3 | 2 grade 0 | 11 grade 0 11 grade 1 2 grade 2 | 4 grade 2 1 grade 3 | 2 grade 2 1 grade 3 | 1 grade 3 |
| Regurgitant jet size | 2 grade 0 15 grade 1 3 grade 2 6 grade 3 7 grade 4 2 grade 5 | 2 grade 0 | 15 grade 1 3 grade 2 5 grade 3 1 grade 4 | 1 grade 3 4 grade 4 | 2 grade 4 1 grade 5 | 1 grade 5 |
| LVIDdN (cm/kg) | 1.53 (1.34–1.81) | 1.49 (1.35–1.64) | 1.47 (1.30–1.56) | 2.02 (1.77–2.12) | 2.10 (2.04–2.24) | 2.30 |
| ESVI (ml/m$^2$) | 21.97 (13.85–33.81) | 16.13 (13.35–18.91) | 19.15 (13.07–26.81) | 39.96 (35.23–42.17) | 38.38 (35.39–44.96) | 53.82 |
| EDVI (ml/m$^2$) | 74.20 (53.43–110.02) | 70.70 (54.00–87.40) | 66.70 (49.59–79.04) | 147.36 (105.51–162.49) | 160.43 (150.94–189.12) | 198.69 |
| LA/Ao | 1.25 (1.12–1.56) | 1.22 (1.08–1.36) | 1.15 (1.10–1.26) | 2.02 (1.82–2.36) | 2.04 (2.03–2.32) | 2.95 |
| E (m/s) | 0.76 (0.64–0.86) | 0.79 | 0.71 (0.59–0.79) | 0.80 (0.71–1.17) | 1.18 (1.14–1.31) | 1.28 |
| FS (%) | 40.00 (37.00–44.00)* | 44.00 (42.00–46.00) | 38.00 (37.00–41.75) | 43.00 (37.50–48.00) | 45.00 (44.50–45.00) | 42.00 |
| MINE score | 26 mild 3 moderate 6 severe | 2 mild | 22 mild 2 moderate | 2 mild 1 moderate 2 severe | 3 severe | 1 severe |

E = E wave velocity, ESVI = end-systolic volume index, EDVI = end-diastolic volume index, FS = fractional shortening, LA/Ao = left atrial-to-aortic root ratio,

LVIDdN = left ventricular end-diastolic diameter normalized for body weight.

Murmur = systolic heart murmur intensity: 0 = absent, 1 = I-II/VI left apical systolic, or soft, 2 = III-IV/VI bilateral systolic, or moderate and loud, 3 = V-VI/VI bilateral systolic or palpable.

Regurgitant jet size: 0 = absent, 1 = trivial, 2 = trace, 3 = mild, 4 = moderate, 5 = severe.

Sex: F = female, NF = neutered female, M = male.

All data are expressed as median and interquartile range (25th and 75th percentile).

*Values differ significantly at $P < 0.05$ from T0.

0 = absent, for the heart murmur; 0 = absent, for the mitral regurgitation; mild, for the MINE score. Fig 2 shows the marginal effect of plasma miR-30b-5p expression on each ordinal variable at T0 and T1.

## Discussion

At present, the pathophysiological aspect of the MMVD is not fully elucidated, especially regarding the molecular background of the disease. The CKCS represent a perfect model for studies aimed at filling such a knowledge gap, considering the hereditability of the disease in this breed, the high prevalence, and the fact that in CKCS MMVD can develop at a relatively young age and show a fast progression [3–6,36–38]. Thus, CKCSs have been enrolled in the present study. In dogs, the diagnosis of MMVD is typically based on a combination of two-dimensional and Doppler echocardiographic findings [2]. However, this approach necessarily required specialized equipment and well-trained cardiologists, which may be not always available in all veterinary institutions. Moreover, since echocardiograms are often acquired only after the recognition of a heart murmur on physical examination, such an approach may lead

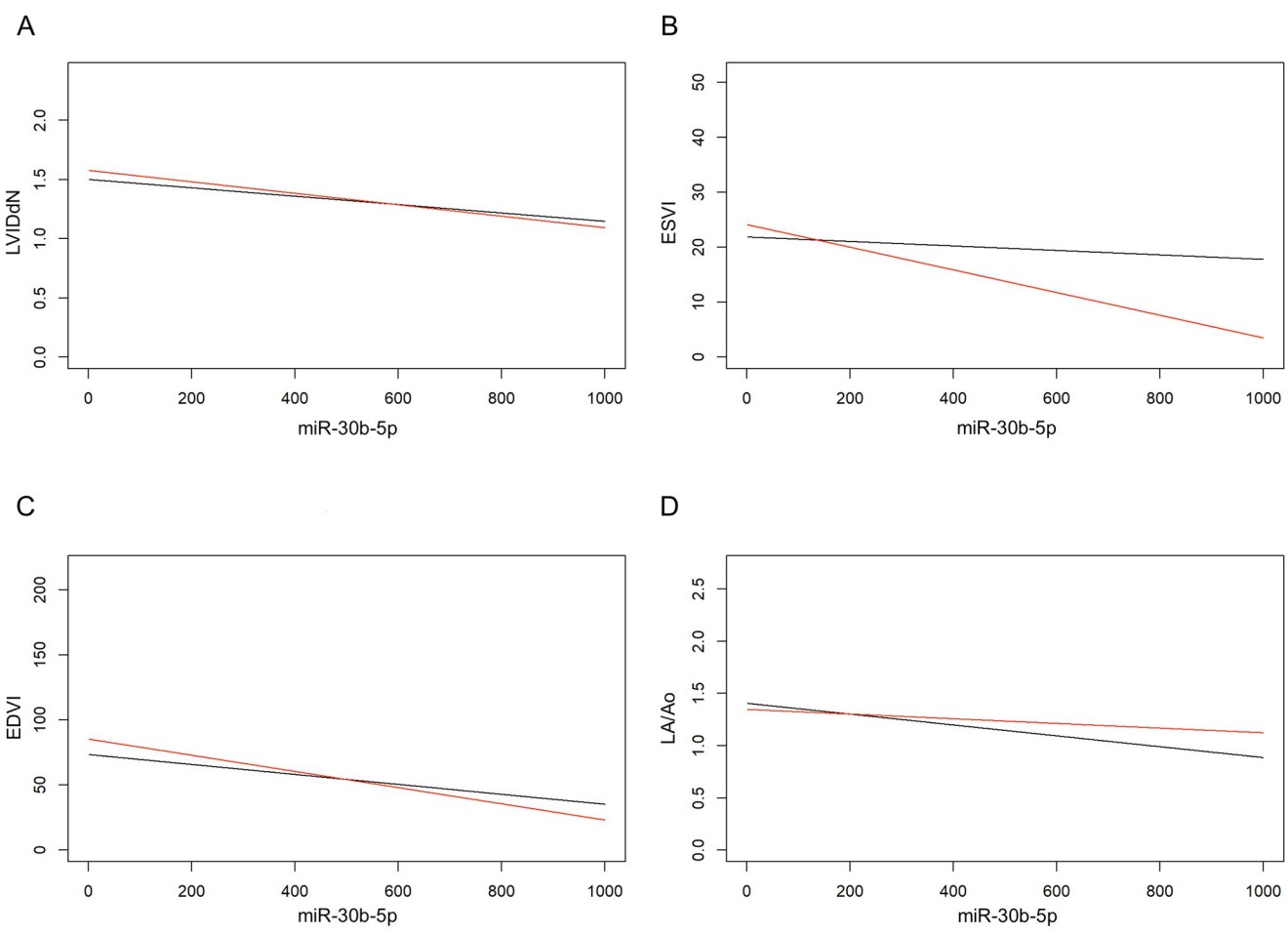

**Fig 1. Marginal effect of plasma miR-30b-5p expression on echocardiographic variables at T0 and T1.** Black line: T0. Red line: T1. A) Marginal effect of miR-30b-5p on LVIDdN. B) Marginal effect of miR-30b-5p on ESVI: The interaction between the two parameters is high since T0 and T1 lines are poorly parallel. C) Marginal effect of miR-30b-5p on EDVI. D) Marginal effect of miR-30b-5p on LA/Ao.

to tardive diagnosis with potentially relevant clinical consequences, especially considering that in some dogs the disease progression may be particularly rapid. Investigation of reliable early biomarkers of MMVD, such miRNAs, could offer additional advantages in the recognition of the disease, with the aim of reaching prompt diagnosis and, consequently, trying to improve therapeutic timing and life expectancy.

A study published by Bagardi et al. [7] conducted on the CKCS recently provided interesting results for miR-30b-5p in ACVIM class B1 CKCSs, demonstrating that miR-30b-5p can discriminate between B1 subjects and healthy dogs, and B1 CKCSs younger than 3 years of age and healthy dogs [7]. Since a heart murmur is uncommon to detect in young patients and echocardiographic changes can be misclassified as normal findings by inexperienced sonographers, miR-30b-5p can support the early diagnosis of MMVD-affected dogs [34]. The results reported in the present pilot study may further strengthen the potential clinical utility of miR-30b-5p measurement in CKCSs affected by MMVD. We achieved this result by investigating, for the first time in veterinary medicine, the relationship between a plasma miRNA and the variation over time of several clinical and echocardiographic parameters in healthy and

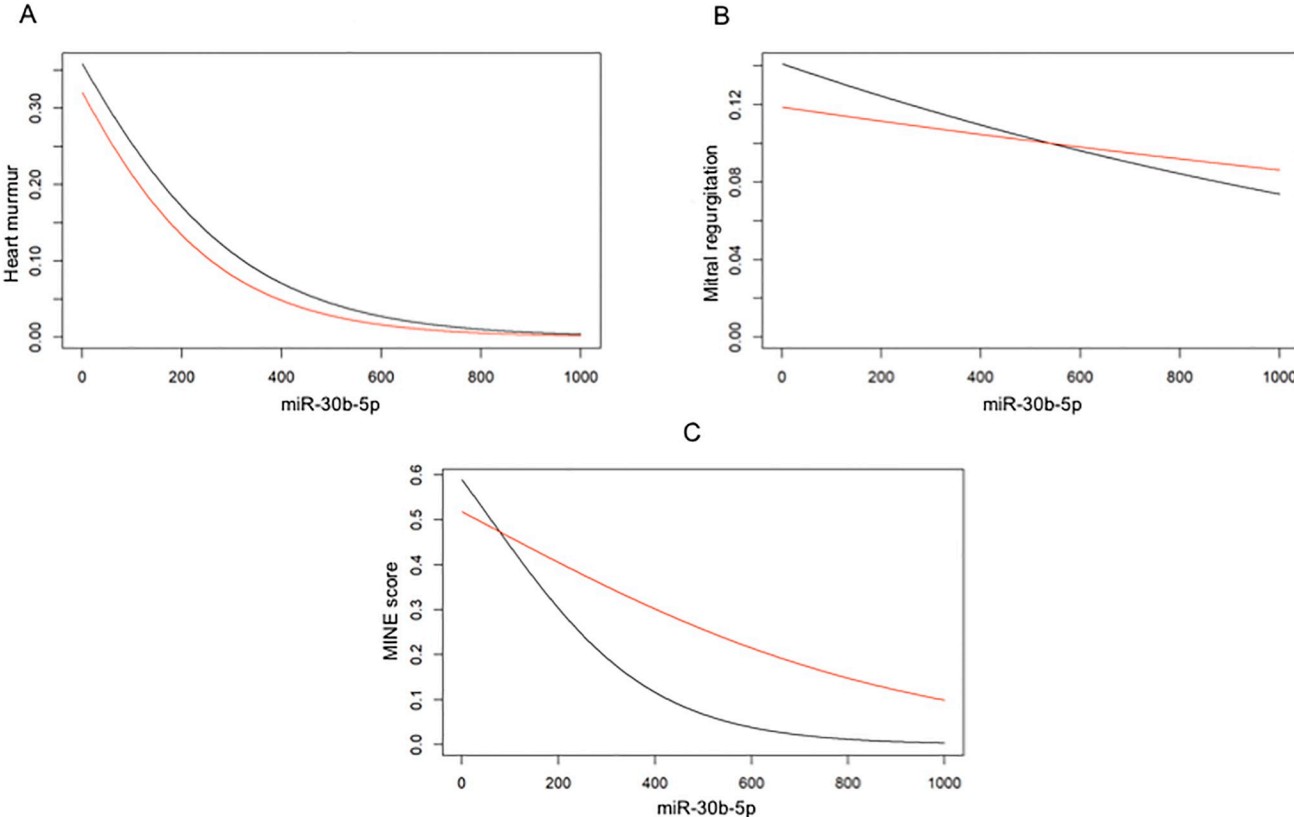

**Fig 2. Marginal effect of plasma miR-30b-5p expression on ordinal variables at T0 and T1.** Black line: T0. Red line: T1. On the y-axis, the probability of the class with the lower value is reported. A) Marginal effect of miR-30b-5p on the presence/intensity of a heart murmur. B) Marginal effect of miR-30b-5p on the mitral regurgitant jet size. C) Marginal effect of miR-30b-5p on the MINE score.

MMVD-affected CKCSs belonging to different ACVIM classes. Specifically, our results highlight a significant and negative marginal effect of miR-30b-5p on some important echocardiographic variables, since high levels of plasma miR-30b-5p corresponded to lower LVIDdN, ESVI, EDVI, and LA/Ao. As these parameters are pivotal to the MMVD diagnosis and monitoring because their increase over time reflects the progression of the disease [30,39–41], we could speculate that higher miR-30b-5p expression is related to an echocardiographically more stable form of MMVD over time in affected CKCSs.

These results are partially supported by the IPW analysis on the miRNA expression and the intensity of a heart murmur, the mitral regurgitation severity, and the MINE score. Indeed, although not statistically significant, these results suggest a positive marginal effect of miR-30b-5p on these three variables. High level of circulating miR-30b-5p corresponded to a higher probability of the class with the lower value (i.e., absence of a heart murmur, absence of a mitral regurgitation, and a mild MINE score). It is possible to hypothesize that the limited sample size of our study and the division of enrolled dogs into several classes of severity for each of these variables prevented us from obtaining statistically significant results. For this reason, studies enrolling a larger number of affected dogs are needed for validating and furtherly expand these results. Furthermore, it is fair to assume that the non-statistically significant results for the MINE score could be due to the fact that two of the parameters on which the score is calculated (i.e., E and FS%) showed no statistical significance on the IPW analysis.

Lastly, our results show a non-significant marginal effect of the elapsed time between T0 and T1 on all the examined variables. This is a somewhat surprising result, since MMVD is a disease that progresses over time. Our main hypothesis is that the lack of homogeneity of the follow-up period and its relatively limited extension (it ranged from 6 to 24 months) may have influenced the IPW analysis providing a hardly realistic result.

This pilot study presents some limitations, such as the small sample size and the limited number of subjects for each ACVIM class. Furthermore, the miR-30b-5p was evaluated only at T0 and not at T1. However, to fill such a gap, we are continuing to collect blood samples and data during the breed screenings, with the goal of setting up a larger prospective study that will include the evaluation of different miRNAs at established follow-up periods and, thus, trying to consolidate the preliminary findings from the present report. Another limitation was the variability of the follow-up, that did not always allow to have clinical and echocardiographic data associated to a fixed timing for each patient. The standardization of T1, and possibly subsequent timings, compared to T0 could clarify the variability of the biomarker due to the progression of the MMVD in this breed.

## Conclusions

In conclusion, this pilot study allows to identify a relationship between high levels of miR-30b-5p and forms of MMVD that appear echocardiographically more stable over time in the CKCS.

## Supporting information

**S1 Table. Estimates of miR-30b-5p marginal effects for continuous and ordinal variables.** E = E wave velocity, EDVI = end-diastolic volume index, ESVI = end-systolic volume index, FS = fractional shortening, LA/Ao = left atrial-to-aortic root ratio, LVIDdN = left ventricular end-diastolic diameter normalized for body weight, Murmur = systolic heart murmur intensity. 95% Lower limit and 95% Upper limit: Lower and upper limits of 95% confidence interval for the mean marginal effect.
(DOCX)

## Acknowledgments

The Authors are thankful to the dog owners and breeders who trustingly and wholeheartedly participated to this research work.

## Author Contributions

**Conceptualization:** Cristina Lecchi, Mara Bagardi, Paola G. Brambilla.

**Data curation:** Sara Ghilardi, Mara Bagardi, Celeste Franco.

**Formal analysis:** Cristina Lecchi, Mara Bagardi, Fabio M. Colombo.

**Investigation:** Cristina Lecchi.

**Methodology:** Sara Ghilardi, Mara Bagardi.

**Project administration:** Paola G. Brambilla.

**Software:** Fabio M. Colombo.

**Supervision:** Paola G. Brambilla.

**Validation:** Giovanni Romito, Paola G. Brambilla.

**Writing – original draft:** Sara Ghilardi, Mara Bagardi.

**Writing – review & editing:** Sara Ghilardi, Cristina Lecchi, Mara Bagardi, Giovanni Romito, Michele Polli, Paola G. Brambilla.

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
