## [Decision Letter · Decision Letter 0]

25 Oct 2022

PONE-D-22-24617Preliminary prospective study on the predictive significance of plasma miR-30b-5p through the study of echocardiographic modifications in Cavalier King Charles Spaniels affected by different stages of myxomatous mitral valve disease: the PRIME studyPLOS ONE

Dear Dr. BAGARDI,

Thank you for submitting your manuscript to PLOS ONE. After careful consideration, we feel that it has merit but does not fully meet PLOS ONE’s publication criteria as it currently stands. Therefore, we invite you to submit a revised version of the manuscript that addresses the points raised during the review process.

ACADEMIC EDITOR: All issues highlight by reviewers need a careful revision. Moreover, the authors should better highlight the difference compared to previous publication (Bagardi M, Ghilardi S, Zamarian V, Ceciliani F, Brambilla PG, Lecchi C . PloS one 2022; 17: e0266208) both in the title and in the text. This last issue is relevant in order to avoid redundancy.

We look forward to receiving your revised manuscript.

Kind regards,

Vincenzo Lionetti, M.D., PhD

Academic Editor

PLOS ONE

Journal Requirements:

Reviewers' comments:

Reviewer's Responses to Questions

**Comments to the Author**

1. Is the manuscript technically sound, and do the data support the conclusions?

Reviewer #1: Yes

Reviewer #2: Yes

2. Has the statistical analysis been performed appropriately and rigorously? 

Reviewer #1: Yes

Reviewer #2: Yes

3. Have the authors made all data underlying the findings in their manuscript fully available?

Reviewer #1: Yes

Reviewer #2: Yes

4. Is the manuscript presented in an intelligible fashion and written in standard English?

Reviewer #1: Yes

Reviewer #2: No

5. Review Comments to the Author

Reviewer #1: The article is well written. Patient selection is made adequately, even if the echocardiographic parameters used have been limited. Another limitation, already reported by the authors, is the small number of dogs enrolled. The small number did not allow to obtain statistically significant results, so this is configured as a preliminary study and differs very little from the study already published by the same group "Bagardi M, Ghilardi S, Zamarian V, Ceciliani F, Brambilla PG, Lecchi C . Circulating MiR377 30b-5p is upregulated in Cavalier King Charles Spaniels affected by early myxomatous mitral

378 valve disease. PloS one 2022; 17: e0266208. "However, it is useful as an attempt to correlate a biomarker specifically" miR-30b-5p "to mitral pathology in Cavalier King Charles Spaniels affected by

4 different stages of myxomatous mitral valve disease. It is imperative to increase the number of dogs involved in the study to increase the weight of this research. Being able to diagnose mitral disease in Cavalier King Charles Spaniels early, in order to focus attention on dogs that show a greater genetic predisposition and on these patients to start treatment early, could be a useful aid for the clinician and increase quality. life expectancy and life expectancy of the treated dog.

Reviewer #2: The work of Ghilardi et al. aims to highlight a microRNA as a predictive biomarker in plasma for the progression of myxomatous mitral valve disease. The scope of this work is of extreme interest in the study of cardiovascular diseases in order to define a pathology not only in relation of macroscopic parameters but in offering an increasingly precise description of it, relating to molecular aspects and pathological evolution overtime. The results obtained from those type of studies still remain fundamental for future interventions.

I have not suggestions about the strategy used by the authors for the correlation analyses and classification. I found it clear and well executed. The bibliography is consistent.

I have a concern about the new message of this paper because is a follow-up of their previous work (reference 7). Reading the text the flow appears very similar to the previous one. In order to highlight the importance of the work I suggest to add:

-a brief explanation, more incisive, why the miR-30b was selected among the most 3 deregulated microRNAs you highlighted in the previous work. For my knowledge, the selection only based on the highest value of abundance is not sufficient, even the other 2 microRNAs could show the same trend in this model and be predictive. Would be very interesting seeing if the shown results are exclusive for miR-30.

In the introduction, line 94-95, authors report a study on exosomes “This led to the hypothesis that exosomal miRNA expression-level changes may be more 95 specific to the disease status than total plasma miRNAs [21]”. I suggest to discuss the point because is not clear the reason to use plasma if there is a more precise method to describe something.

Corrections.

I often found an improper use of the comma in the text, please revise it.

Please see my suggestions below:

• In the line 51 please delete the comma in “dogs, and a frequent cause of heart failure”

• In the line 66 please add a scientific context in the sentence and a relative citation: “In recent years, the study of microRNAs (miRNAs) increased.”

For example: -In recent years, the study of microRNAs (miRNAs) as predictive biomarkers for … increased”

• In the line 67 please delete a comma and correct the sentence in: “found in plants, animals and some viruses,”

• In the line 72 same, the sentence is not an aside, please correct in: “…of studies on circulating miRNAs is currently brief and few of them…”

• Same in line 104: “concentration of miR-1-3p, miR30b-5p and miR-128-3p in…”; in line 136, 147.

6. PLOS authors have the option to publish the peer review history of their article (what does this mean?). If published, this will include your full peer review and any attached files.

Reviewer #1: No

Reviewer #2: No

---

## [Author Response · Author response to Decision Letter 0]

25 Nov 2022

Reviewers' comments:

The Authors thank the Editor and the Reviewers for their meticulous revision of our work. We are delighted by their positive comments and the appreciation of the manuscript. We think that our work is improved thanks to their suggestions. Thank you very much. 

Best regards.

Reviewer #1: The article is well written. Patient selection is made adequately, even if the echocardiographic parameters used have been limited. Another limitation, already reported by the authors, is the small number of dogs enrolled. The small number did not allow to obtain statistically significant results, so this is configured as a preliminary study and differs very little from the study already published by the same group "Bagardi M, Ghilardi S, Zamarian V, Ceciliani F, Brambilla PG, Lecchi C . Circulating MiR377 30b-5p is upregulated in Cavalier King Charles Spaniels affected by early myxomatous mitral valve disease. PloS one 2022; 17: e0266208. "However, it is useful as an attempt to correlate a biomarker specifically" miR-30b-5p "to mitral pathology in Cavalier King Charles Spaniels affected by different stages of myxomatous mitral valve disease. It is imperative to increase the number of dogs involved in the study to increase the weight of this research. Being able to diagnose mitral disease in Cavalier King Charles Spaniels early, in order to focus attention on dogs that show a greater genetic predisposition and on these patients to start treatment early, could be a useful aid for the clinician and increase quality and life expectancy of the treated dog.

Thank you very much for your comment and your appreciation of our work. We are really pleased about it. As you said, this is a preliminary study: we conducted the reported analysis on a limited number of patients because they were at our disposal at the time of the research. We have added the therm “pilot” in the title and in the material and method section to be more consistent. We thought that the obtained results may be of interest and that is why we felt encouraged to report them. In the meanwhile, we continued recruiting new subjects, so our purpose is to extend the analysis at this larger population in the near future. 

Reviewer #2: The work of Ghilardi et al. aims to highlight a microRNA as a predictive biomarker in plasma for the progression of myxomatous mitral valve disease. The scope of this work is of extreme interest in the study of cardiovascular diseases in order to define a pathology not only in relation of macroscopic parameters but in offering an increasingly precise description of it, relating to molecular aspects and pathological evolution overtime. The results obtained from those type of studies still remain fundamental for future interventions.

I have not suggestions about the strategy used by the authors for the correlation analyses and classification. I found it clear and well executed. The bibliography is consistent.

I have a concern about the new message of this paper because is a follow-up of their previous work (reference 7). Reading the text the flow appears very similar to the previous one. In order to highlight the importance of the work I suggest to add:

-a brief explanation, more incisive, why the miR-30b was selected among the most 3 deregulated microRNAs you highlighted in the previous work. For my knowledge, the selection only based on the highest value of abundance is not sufficient, even the other 2 microRNAs could show the same trend in this model and be predictive. Would be very interesting seeing if the shown results are exclusive for miR-30.

Thank you very much for your comments. The choice to investigate exclusively the miR-30b-5p has been dictated by the promising findings reported in literature in another breed affected by myxomatous mitral valve disease – the Dachshund (Reference 18 in the new version of the Manuscript) – and the findings of our work on the CKCS (Reference 7). In fact, it is not only because of the different abundance of the miR-30b-5p in ACVIM classes A and B1 that we chose to select it for our analyses: we eliminated the other two miRNAs because the results of that work showed no statistical significance in the CKCS plasma. Obviously we are aware that only miR-30b-5p may not be clinically enough and for this reason, as a next step to our research, we are now working on the sequencing of the microRNAs of our population of CKCSs to identify a pool of miRNAs that is unique and specific for the CKCS. Once this is done, we could better understand which miRNAs are up- or downregulated in this breed according to the various stages of MMVD. By far, in fact, we based our research work on the findings reported in literature by other groups. 

In the introduction, line 94-95, authors report a study on exosomes “This led to the hypothesis that exosomal miRNA expression-level changes may be more specific to the disease status than total plasma miRNAs [21]”. I suggest to discuss the point because is not clear the reason to use plasma if there is a more precise method to describe something.

Thank you, we included a sentence in the text to explain why the use of exosomal miRNAs, although promising, is not suggested, yet, and I also included the proper reference. I hope this brief explanation will clear your doubt about it. 

Corrections.

I often found an improper use of the comma in the text, please revise it.

Please see my suggestions below:

• In the line 51 please delete the comma in “dogs, and a frequent cause of heart failure”

• In the line 67 please delete a comma and correct the sentence in: “found in plants, animals and some viruses,”

• In the line 72 same, the sentence is not an aside, please correct in: “…of studies on circulating miRNAs is currently brief and few of them…”

• Same in line 104: “concentration of miR-1-3p, miR30b-5p and miR-128-3p in…”; in line 136, 147.

Thank you, we have modified the text as you suggested.

• In the line 66 please add a scientific context in the sentence and a relative citation: “In recent years, the study of microRNAs (miRNAs) increased.”

For example: -In recent years, the study of microRNAs (miRNAs) as predictive biomarkers for … increased”

Thank you, we better explained the sentence and give it the proper reference.

---

## [Decision Letter · Decision Letter 1]

12 Dec 2022

Prospective pilot study on the predictive significance of plasma miR-30b-5p through the study of echocardiographic modifications in Cavalier King Charles Spaniels affected by different stages of myxomatous mitral valve disease: the PRIME study

PONE-D-22-24617R1

Dear Dr. BAGARDI,

We’re pleased to inform you that your manuscript has been judged scientifically suitable for publication and will be formally accepted for publication once it meets all outstanding technical requirements.

Kind regards,

Vincenzo Lionetti, M.D., PhD

Academic Editor

PLOS ONE

Additional Editor Comments (optional):

Reviewers' comments:

Reviewer's Responses to Questions

**Comments to the Author**

1. If the authors have adequately addressed your comments raised in a previous round of review and you feel that this manuscript is now acceptable for publication, you may indicate that here to bypass the “Comments to the Author” section, enter your conflict of interest statement in the “Confidential to Editor” section, and submit your "Accept" recommendation.

Reviewer #1: All comments have been addressed

Reviewer #2: All comments have been addressed

2. Is the manuscript technically sound, and do the data support the conclusions?

Reviewer #1: Yes

Reviewer #2: (No Response)

3. Has the statistical analysis been performed appropriately and rigorously? 

Reviewer #1: Yes

Reviewer #2: (No Response)

4. Have the authors made all data underlying the findings in their manuscript fully available?

Reviewer #1: Yes

Reviewer #2: (No Response)

5. Is the manuscript presented in an intelligible fashion and written in standard English?

Reviewer #1: Yes

Reviewer #2: (No Response)

6. Review Comments to the Author

Reviewer #1: The authors added the word "pilot" in the title and material

and the method section to be more consistent; furthermore, they stated that they will continue to recruit new subjects to extend the analysis to this one

larger population in the near future.

The results obtained, although similar to the previous work of the same group (Bagardi M, Ghilardi S, Zamarian V, Ceciliani F, Brambilla PG, Lecchi C. Circulating MiR385 30b-5p is upregulated in Cavalier King Charles Spaniels affected by early myxomatous mitral valve disease. PloS one 2022;17:e0266208), could

be of interest.

Reviewer #2: (No Response)

7. PLOS authors have the option to publish the peer review history of their article (what does this mean?). If published, this will include your full peer review and any attached files.

Reviewer #1: No

Reviewer #2: No

---

## [Editor Report · Acceptance letter]

15 Dec 2022

PONE-D-22-24617R1 

Prospective pilot study on the predictive significance of plasma miR-30b-5p through the study of echocardiographic modifications in Cavalier King Charles Spaniels affected by different stages of myxomatous mitral valve disease: the PRIME study 

Dear Dr. Bagardi:

I'm pleased to inform you that your manuscript has been deemed suitable for publication in PLOS ONE. Congratulations! Your manuscript is now with our production department. 

Kind regards, 

on behalf of

Prof. Vincenzo Lionetti 

Academic Editor

PLOS ONE